# Feasibility of Polyclonal Avian Immunoglobulins (IgY) as Prophylaxis against Human Norovirus Infection

**DOI:** 10.3390/v14112371

**Published:** 2022-10-27

**Authors:** Chad Artman, Nnebuefe Idegwu, Kyle D. Brumfield, Ken Lai, Shirley Hauta, Darryl Falzarano, Viviana Parreño, Lijuan Yuan, James D. Geyer, Julius G. Goepp

**Affiliations:** 1Scaled Microbiomics, LLC, Hagerstown, MD 21740, USA; 2Maryland Pathogen Research Institute, University of Maryland, College Park Campus, College Park, MD 20742, USA; 3University of Maryland Institute for Advanced Computer Studies, University of Maryland, College Park Campus, College Park, MD 20742, USA; 4Vaccine and Infectious Disease Organization, University of Saskatchewan, Saskatoon, SK S7N 5E3, Canada; 5Department of Veterinary Microbiology, Western College of Veterinary Medicine, University of Saskatchewan, Saskatoon, SK S7N 5B4, Canada; 6Department of Biomedical Sciences and Pathobiology, Virginia-Maryland College of Veterinary Medicine, Virginia Polytechnic Institute and State University, Blacksburg, VA 24061, USA; 7INCUINTA, IVIT, National Institute of Agricultural Technology (INTA, Argentina), Buenos Aires 1712, Argentina; 8Institute for Rural Health Research, College of Community Health Science, University of Alabama, Tuscaloosa, AL 35487, USA

**Keywords:** outbreak prevention and control, norovirus, calicivirus, foodborne disease, gastroenteritis outbreaks, IgY, passive immunotherapy

## Abstract

Background: Human norovirus (HuNoV) is the leading viral cause of diarrhea, with GII.4 as the predominant genotype of HuNoV outbreaks globally. However, new genogroup variants emerge periodically, complicating the development of anti-HuNoV vaccines; other prophylactic or therapeutic medications specifically for HuNoV disease are lacking. Passive immunization using oral anti-HuNoV antibodies may be a rational alternative. Here, we explore the feasibility of using avian immunoglobulins (IgY) for preventing HuNoV infection in vitro in a human intestinal enteroid (HIE) model. Methods: Hens were immunized with virus-like particles (VLP) of a GII.4 HuNoV strain (GII.4/CHDC2094/1974/US) by intramuscular injection. The resulting IgY was evaluated for inhibition of binding to histo-blood group antigens (HBGA) and viral neutralization against representative GII.4 and GII.6 clinical isolates, using an HIE model. Results: IgY titers were detected by three weeks following initial immunization, persisting at levels of 1:2^21^ (1:2,097,152) from 9 weeks to 23 weeks. Anti-HuNoV IgY significantly (*p* < 0.05) blocked VLP adhesion to HBGA up to 1:12,048 dilution (0.005 mg/mL), and significantly (*p* < 0.05) inhibited replication of HuNoV GII.4[P16] Sydney 2012 in HIEs up to 1:128 dilution (0.08 mg/mL). Neutralization was not detected against genotype GII.6. Conclusions: We demonstrate the feasibility of IgY for preventing infection of HIE by HuNoV GII.4. Clinical preparations should cover multiple circulating HuNoV genotypes for comprehensive effects. Plans for animal studies are underway.

## 1. Introduction

Enteric diseases causing severe diarrhea threaten survival of children and are a source of considerable morbidity in low- and middle-income countries (LMIC) [1]. Human noroviruses (HuNoVs), a group of single-stranded RNA, non-enveloped viruses in the family *Caliciviridae*, are the predominant viral pathogens associated with acute gastroenteritis in humans [2,3]. HuNoVs are common etiopathogens in travelers’ diarrhea (TD), affecting children and adults traveling from industrialized nations to LMIC, as well as military and diplomatic personnel stationed in endemic regions [4,5]. HuNoV is also a common agent of outbreaks in long-term care facilities, contributing to excess morbidity and mortality among older adults [6]. 

Importantly, chronic HuNoV infections can cause gastroenteritis and intestinal barrier dysfunction in both primary and acquired immunodeficiency states, which include patients undergoing immunosuppressive therapies related to hematopoietic stem cell and solid organ transplants, those with human immunodeficiency virus infection, and those with cancer or undergoing cancer treatment [7]. Although the overall incidence of HuNoV disease in hospital and community settings remains unclear, studies in hematopoietic stem cell transplant recipients suggest an 18% infection rate over one year, and a two-year survey of kidney transplant recipients reported a 17% chronic HuNoV infection rate [7,8,9,10].

To establish infection, HuNoV must first adhere to mucosal epithelial cells in the duodenum or upper jejunum [11,12]. Protruding (P) domains of the viral capsid mediate HuNoV binding to histo-blood group antigens (HBGAs), resulting in cellular invasion and virus replication, and inducing the physical symptoms associated with acute gastroenteritis [13,14]. Efforts to develop vaccines focus overwhelmingly on preventing or interrupting adhesion of the P-domain to HBGA receptors [15]. Hence, viral capsid domains have been targets of HuNoV vaccine candidates [3]. However, diversity among HuNoV variants, including mutations within viral capsid domains, and the emergence of newer genotypes in genogroups I and II (GI and GII, respectively) has complicated vaccine development [2,16]. Currently, no licensed vaccine against HuNoV is available, though several are in early-stage clinical trials [2,16].

An appealing alternative to active immunization by vaccination is to provide passive immunity by delivering protective antibodies directly to the intestinal sites of HuNoV infection. Polyclonal immunoglobulin Y (IgY), the primary circulating antibody of avians, offers certain advantages over mammalian antibodies. For example, therapeutic IgY candidates can be obtained by immunization of laying hens with target antigens, subsequent egg collection and non-invasive harvesting of IgY [17]. Antibody obtention is fast, simple and cost-effective and aligned with the 3Rs (Replacement, Reduction, and Refinement) of animal welfare [18].

IgY passive immunotherapy has been demonstrated to be effective against a few other enteropathogens, including human rotavirus in animal models [19] and in human infants [20], and in treatment of *Helicobacter pylori* [21]. No published studies have yet demonstrated effectiveness of IgY as an immunoprophylactic in humans, to our knowledge.

Given the absence of available vaccines for HuNoV, the high mortality associated with diarrhea in LMIC, the elderly, and immunocompromised people with chronic HuNoV infection, and evidence that passive immunization with IgY is effective in animal models [22], we set out to evaluate the feasibility of producing IgY targeting HuNoV as a potential diarrheal prophylactic in humans, including evaluation in a human intestinal enteroid (HIE) model. Results demonstrated the feasibility of preparing IgY for further evaluation as potential prophylaxis against HuNoV.

## 2. Materials and Methods

### 2.1. Human Norovirus-like Particles and IgY Targets

HuNoV genotype GII.4 remains the predominant genotype in HuNoV outbreaks globally [2,3], and hence the target of this investigation. HuNoV immunogens used for IgY production were recombinant HuNoV-like particles (HuNoVLP) comprising the capsid protein VP1 of HuNoV genotype GII.4 (HuNoV GII.4/CHDC2094/1974/US; GenBank accession number ACT76142.1; hereafter CHDC2094), produced commercially (The Native Antigen Company, Oxfordshire, UK) in mammalian HEK293 cells to form particles without the non-structural proteins or genomic material.

### 2.2. Immunization of Laying Hens

Eight commercial White Rock and Rhode Island cross-bred, sex-linked hens (Pinola Hatchery, Shippensburg, PA, USA) were housed in a purpose-built henhouse permitting segregation of paired hens. Hens were acclimated for two weeks prior to immunization at ambient temperatures on a 12 h light/dark cycle on ad libitum water and a commercial diet (Martin’s Layer Mash 16%, Martin’s Elevator, Inc., Hagerstown, MD, USA). Protocols for hen maintenance and immunization were approved by the Scaled Microbiomics, LLC Animal Use and Care Committee (approval number 19-01-TD).

HuNoVLP were diluted in phosphate-buffered saline (PBS) and emulsified with Montanide ISA 70 VG adjuvant (Seppic, Inc., La Garenne-Colombes, France) in a 7:3 *v/v* ratio using a high-shear blender. The immunogen having a final concentration of 100 µg of HuNoVLP per ml was filter-sterilized using a 0.2 µm pore-size polyethersulfone filter membrane (VWR International, Radnor, PA, USA). Sterility was confirmed by the absence of visual growth after inoculating 25 µL of each vaccine mixture into fastidious BBL™ Schaedler broth with Vitamin K_1_ (Becton Dickinson, Sparks, MD, USA) and incubating for 48 h without aeration at 37 °C. On day 1 of hen immunizations, two pairs of separately housed hens were injected with 0.5 mL in each breast muscle, delivering a total of 100 µg of HuNoVLP per hen. Booster injections were administered in an identical fashion on days 14 and 28 of hen immunizations. Two additional hens, designated “sham injected”, received 0.5 mL of PBS and adjuvant only, prepared as previously described without HuNoVLP, in each breast during immunizations (days 1, 14, and 28). A final hen pair was used for control and received no immunizations, designated “unimmunized”.

### 2.3. IgY Extraction and Concentration

Two eggs were collected weekly from each hen pair beginning one day prior to the first immunization. IgY was extracted from yolks using polyethylene glycol (PEG), as described elsewhere [17], with the following modifications. Briefly, yolks were pooled, and lipid content was removed by centrifugation (13,000× *g* for 20 min at 4 °C) using PEG 6000 at consecutively increasing concentrations (3.5, 8.5, and 12 % *w*/*v*; Alfa Aesar, Haverhill, MA, USA). The resulting precipitate was resuspended in PBS and dialyzed against sodium chloride (0.1% *w*/*v*) for 16 h and PBS for an additional three hours using Spectra/Por 4 standard regenerated cellulose dialysis tubing (12–14 kD; Spectrum Laboratories, Inc., Rancho Dominguez, CA, USA). The resulting water-soluble fraction (WSF) containing IgY was stored at −20 °C until further analysis (<two weeks).

Total protein concentrations of WSF were determined by bicinchoninic acid (BCA) method kit (Thermo Fisher Scientific, Rockford, IL, USA), following the manufacturer’s specifications. Absorbance values were read at 490 nm using a THERMOmax microplate reader (Molecular Devices, Sunnyvale, CA, USA), and the standard curve showed linear behavior (R^2^ = 0.99) over seven serial 1:2 dilutions (0.06–2 mg/mL) of the bovine serum albumin protein standard set (Thermo Fisher Scientific, Waltham, MA, USA).

### 2.4. Sodium Dodecyl Sulfate-Polyacrylamide Gel Electrophoresis (SDS-PAGE)

To determine the purity of yolk-derived IgY, SDS-PAGE was conducted under reducing conditions using 12% polyacrylamide gel (NuSep Inc., Germantown, MD, USA) with a Novex Mini-Cell (Invitrogen, Carlsbad, CA, USA). Briefly, purified WSF samples were diluted 1:10 in PBS, mixed with an equal volume of sample buffer, and denatured for 5 min at 100 °C. A total of 20 µL of sample/buffer mixture was loaded into each well, and protein bands were visualized with Protein Fixative (Ward’s Science, Rochester, NY, USA), as recommended by the manufacturer. 

### 2.5. Enzyme-Linked Immunosorbent Assay: IgY in HuNoVLP-Immunized Hens’ Eggs

IgY titers against the CHDC2094 antigen were measured by indirect noncompetitive ELISA, as reported previously [23,24,25,26], with slight modifications. Briefly, 96-well flat-bottom microtiter plates were coated with 400 ng of HuNoVLP antigen. Plates were blocked with 5% nonfat milk and incubated with three serial 2-fold dilutions of IgY for one hour at room temperature (23–25 °C). Bound anti-HuNoVLP IgY was detected by horseradish peroxidase (HRP)-conjugated goat anti-chicken IgY diluted 1:2500 *v*/*v* (ImmunoReagents, Inc., Raleigh, NC, USA). Plates were washed five times using commercial ELISA wash buffer (Thermo Fisher Scientific, Waltham, MA, USA) and visualized by adding 100 μL 3,3′-5,5′-tetramethylbenzidine (TMB; VWR International, Radnor, PA, USA), followed by 100 μL 2N sulfuric acid stop solution. Optical density (OD) was measured on a THERMOmax microplate reader (Molecular Devices, Sunnyvale, CA, USA) at 450 nm (OD_450_). Antigen-specific IgY titer was defined as the maximum dilution of the sample with an OD_450_ value that was 2.1 times the unimmunized control.

### 2.6. HuNoVLP-Targeted IgY: HuNoVLP Adherence Inhibition Assays

Pig gastric mucin (PGM), Type III with HBGA type A, Ley and H2, (Sigma Aldrich, St. Louis, MO, USA) was used in an antibody-blocking assay, as described previously [27]. Briefly, U-bottom 96-well vinyl microtiter plates (Thermo Fisher Scientific, Rockford, IL, USA) were coated with 1 μg of PGM in 100 μL of PBS per well for 4 h at room temperature. Plates were blocked overnight at 4 °C in 5% skim milk in 0.05% Tween 20-PBS. 200 µg HuNoVLPs were pre-treated for 1 h at room temperature with five decreasing serial twofold dilutions of anti-CHDC2094 IgY, beginning with a starting concentration of 78 μg/mL. A total of 100 μL HuNoVLP-IgY mixture was transferred to the PGM-coated plates and incubated for 1 h at 37 °C. Plates were washed three times with 0.05% Tween 20-PBS, and bound HuNoVLPs were detected using a diluted (1:10,000) monoclonal anti-GII.4 VP1 VLP mouse IgG targeting the shell domain of the major capsid protein of GII.4 HuNoV (LifeSpan BioSciences Inc., Seattle, Washington, USA), following incubation for 1 h at 37 °C. Plates were washed a second time as mentioned and incubated with goat anti-mouse IgG-HRP conjugated antibodies (Azure Biosystems, Dublin, CA, USA) at a 1:2000 dilution for 1 h at 37 °C. Following a final series of five washes, the assay was developed with commercial TMB substrate (VWR International, Radnor, PA, USA) using 100 μL/well. The OD was measured at 650 nm (OD_650_) using a THERMOmax microplate reader (Molecular Devices, Sunnyvale, CA, USA), every 5 min, for up to 40 min or until linearity was established. Percent binding was expressed as percent absorbance of the uninhibited blank control and confirmed by comparison to unimmunized control IgY. HBGA assays are presented as box and whisker plots of inhibition of HuNoVLP adhesion to HBGA antigens in a cell-free system. Boxes represent interquartile range (IQR) with median shown as the center bar of each sample group (N = 3 replicates per group). Whiskers represent 1.5 times the IQR. *p*-value, by two-sample *t*-test method, and 95% confidence interval (CI) was calculated using the R software package EnvStats (v.2.3.1) [28].

### 2.7. HuNoV Neutralization in Human Intestinal Enteroids

To evaluate the impact of the anti-GII.4 IgY against active HuNoV GII.4, a HIE model was employed at the Vaccine and Infectious Disease Organization (VIDO, University of Saskatchewan, Saskatoon, SK, Canada) under the support of the National Institute for Allergy and Infectious Diseases (NIAID) [29,30]. 

#### 2.7.1. Human Jejunal Enteroid Culture

All human jejunal enteroid culture assays were conducted using HIE jejunal J2 cells (Baylor College of Medicine, Houston, TX, USA). Three-dimensional HIEs J2 were maintained as previously described [30]. For norovirus replication assays two-dimensional cells were generated. The two-dimensional HIE J2 cells were seeded onto collagen IV (Sigma-Aldrich Canada, Oakville, ON, Canada)-coated 96-well plates at a density of 9 × 10^4^ cells/well. Cells were grown on Proliferation-IntestiCult medium (Stem Cell Technology, Vancouver, BC, Canada) supplemented with 10 µM Y-27632 dihydrochloride (Sigma-Aldrich Canada, Oakville, ON, Canada) at 37 °C, 5% CO_2_ for 24 h. Cells were then grown on Differentiation-IntestiCult medium (StemCell Technologies, Vancouver, BC, Canada) for up to five days until confluent monolayers were observed, as previously described elsewhere [31].

#### 2.7.2. Human Norovirus Strains

For norovirus HIE replication assays, three different HuNoV stool filtrates were used: Two different genotype GII.4 [P16] Sydney isolates collected from patients in Alberta and Saskatchewan, Canada (isolates AB2 and 3241SK, respectively), which have known capsid sequence heterology from the immunizing 1974 strain [32], and a genotype GII.6 collected from a patient in Saskatchewan (isolate 3650SK), to determine if there was any cross-neutralization between GII.4-directed IgY and a GII.6 virus. The dual typing conventional RT-PCR assay method was used to determine the genotype of the virus in the stool filtrate [32]; sequences derived from this method were genotyped using the Centers for Disease Control and Prevention’s CaliciNet [33]. Genotyping primers are included in Table 1. The 10% norovirus stool filtrates contained approximately 1 × 10^7^ genome copies per microliter were previously prepared. For norovirus HIE replication assays, the stool filtrates were diluted 1000-fold in complete media without growth factors (CMGF-) supplemented with 500 µM glycochenodeoxycholic acid (GCDCA) media (Sigma-Aldrich Canada, Oakville, ON, CA), to a viral concentration of 1 × 10^4^ genome copies per microliter. Following a 1 h incubation at 37 °C, 5% CO_2_, the inoculum was removed, cells were washed twice and Differentiation-IntestiCult medium with GCDCA was added. 

Filtrates were subsequently quantified using a modified 50% Tissue Culture Infectious Dose (TCID_50_) assay to measure HuNoV infectivity, as previously described [12] in J2 HIE. This method was chosen because HuNoV does not cause cytopathic effects in HIE, making the classical TCID_50_ (or plaque assay) method inapplicable. Briefly, J2 HIE monolayers were infected as described above in sextuplet with 2-fold serially diluted stool filtrates starting at 1:1000 dilution. Monolayers were washed twice and overlaid with media as above. Cells were harvested 24 h later for RT-qPCR. Wells were scored as positive as a result of a copy number increase of norovirus genome of at least 10-fold as compared to the 1 h post-infection control, and the resulting positive or negative score was used to calculate the TCID_50_ using the Spearman–Karber method as described in [34]. The 10% stool filtrate for AB-2 (GII.4), 3241SK (GII.4) and 3650SK (GII.6) contained 2.25 × 10^5^ TCID_50_/mL, 7.98 × 10^4^ TCID_50_/mL and 7.96 × 10^4^ TCID_50_/mL, respectively.

#### 2.7.3. Anti-HuNoV Activity of IgY Antibodies on HIE Cells

Using a protocol based upon prior work by Alvarado et al. [37], anti-HuNoV activity of IgY antibodies was tested initially over a broad range, i.e., by 10-fold serial dilutions up to 1:1000 against AB2. Additionally, tested was IgY raised against a novel predicted HuNoV receptor binding domain (RBD) [14]. The virus was prepared as described above and added to serially diluted antibodies in a 96-well round bottom plate and incubated for 2 h at 37 °C, 5% CO_2_. Subsequently, the virus/antibody solution was added to washed two-dimensional HIEs, and after one hour incubation at 37 °C, 5% CO_2_, the inoculum was removed, and culture plates were washed 2 times with room-temperature CMGF- and replaced with Differentiation-IntestiCult medium with GCDCA. Plates were then incubated for 24 h. All IgY antibodies were tested undiluted (10 mg/mL) and at a 10-fold concentration in a pilot experiment. Subsequently, eight two-fold serial dilutions of the IgY antibodies were performed. Assays were repeated twice and performed with four technical replicates. Medium alone was used as a control for cell growth, and medium with virus but without antibody was used as a control for infection. 

The model compound 2′-C-methylcytidine (2-CMC, Sigma-Aldrich Canada, Oakville, ON, Canada) was used as an antiviral control. Concentrations of 2′-CMC between 200–400 µM completely inhibited GII.4 norovirus replication in J2 HIE cells (Appendix A). Therefore, 2′-CMC concentration of 200 µM was used in all experiments.

#### 2.7.4. Determination of HuNoV Genome Copies in J2 HIE Cells

HuNoV viral loads in the infected J2 HIE cells were determined by reverse transcriptase quantitative polymerase chain reaction (RT-qPCR) and are reported as genome copies. Briefly, RNA was isolated from cells infected with HuNoV for 24 h using Direct-zol RNA Miniprep kit (Zymo Research, Irvine, CA, USA). RT-qPCR was performed on the RNA samples using the HuNoV primers and probes listed in Table 1.

A standard curve was prepared using the HOV36 RNA Transcript serially diluted from 2 × 10^6^ to 2 × 10^1^ genome copies per reaction (Appendix A) [38]. Synthesis of cDNA was performed using qScript XLT 1-step RT-qPCR Tough Mix (Quantabio, Beverly, MA, USA), which includes ROX passive dye. RT-qPCR was performed on the Applied Biosciences StepOnePlus RT-PCR machine (ThermoFisher, Waltham, MA, USA) using the following protocol: 50 °C (15 min); 95 °C (5 min), and 40 cycles of 95 °C (15 s) to 60 °C (35 s). Quantification cycle values were exported to Excel (Microsoft, Redmon, WA, USA) for further analysis. Under these experimental conditions, the assay has a limit of detection of 2 × 10^2^ genome copies per reaction.

## 3. Results

### 3.1. HuNoVLPs Induce IgY Antibody Response in Hens

A first step in the development of new IgY therapeutic candidates is assessment of the antigen’s safety in the antibody-producing animal, and of the immunogenicity of the antigen in that context. We observed no adverse effects of any immunizations, including local or systemic inflammatory effects, changes in animal weight or number of eggs produced daily before and after the immunization series. ELISA of IgY production in hens immunized with CHDC2094 HuNoVLP showed detectable antibody production by three weeks after the first immunization, achieving post-immunization titers of 2^−^^25^ (1:33,554,432) at six weeks before falling to 2^−^^21^ (1:2,097,152) at nine weeks. Production of anti-HuNoVLP IgY was sustained at or above these titers until at least 23 weeks when the recording period ended (Appendix A). Recent sampling has demonstrated that at nine months post-immunization, titers remain at least 2^−^^17^ (1:131,072). By contrast, ELISA of both unimmunized and sham-immunized hens’ IgY revealed no detectable antigen-specific antibodies (not shown). Therefore, the control condition is hereafter referred to as “Unimmunized IgY”.

Following SDS-PAGE, bands were observed at the molecular weights expected for purified IgY, with a heavy chain at 68 kD and light chain 24 kD, respectively (Appendix A). The concentration of purified IgY in PBS after dialysis was approximately 10 mg/mL, determined by BCA assay. IgY extracted from pooled yolks during weeks 7–11 of egg collection, at a concentration of 10 mg/mL, was used in all subsequent analyses. Taken together these findings demonstrate that HuNoV GII.4 VLP in our selected adjuvant is safe for laying hens, and that sustained, long-term production of high-titer antibodies is possible.

### 3.2. Anti-HuNoVLP IgY Inhibits Cognate VLP Adhesion to Histo-Blood Group Antigens In Vitro

To screen anti-HuNoV GII.4 VLP IgY as a candidate for viral inhibition we evaluated its ability to inhibit binding of the VLP to histo-blood group antigen (HBGA), the cell-surface adhesion point for HuNoV. In an in vitro cell-free system using HBGA as the adhesion target, anti-CHDC2094 IgY significantly (*p* < 0.05; coefficient of variance (CV) 0.013–0.110; Appendix A) inhibited binding of CHDC2094 VLP to the adhesion target, compared with both “No IgY” and “Unimmunized IgY” conditions at dilutions from 1:128 (0.078 mg protein/mL PBS) to 1:2048 (0.005 mg/mL) (Figure 1). Unimmunized IgY demonstrated a significant (*p* < 0.05; CV 0.013–0.77; Appendix A) inhibition of adhesion compared with No IgY at the highest tested concentration (1:128) only, while anti-HuNoVLP IgY produced significantly (*p* < 0.05) greater inhibition than unimmunized IgY at all concentrations. The reduction in anti-adhesion effects by active IgY at higher dilutions of 1:1024 and 1:2048 is significant (*p* < 0.05) compared with that demonstrated at 1:128, indicating a dose-dependent effect. The strong anti-adhesion properties of the tested antibodies suggested that the material was a good candidate for evaluation in a live virus neutralization model.

### 3.3. Anti-HuNoV GII.4/CHDC2094 IgY Neutralizes Live GII.4[P16] Sydney HuNoV in HIE

We next examined the ability of our anti-GII.4 VLP antibodies to prevent viral replication in an established HIE model. The average fold change (log_10_) for the GII.4 (AB2 isolate) viral genome in culture was 2.87 from hour 1 to hour 24. Undiluted (10 mg/mL) and at 1:10 dilution, HuNoVLP IgY completely inhibited virus replication, while at 1:100 or 1:1000 dilution, no neutralizing activity was detected (Appendix A). Unimmunized IgY did not have any neutralization activities, nor did anti-RBD IgY. IgYs were also further concentrated (two-fold) but this did not lead to increased neutralization activity. Therefore, unimmunized IgY was used as the negative control in subsequent experiments. 

In independent two-fold dilution experiments, HuNoVLP IgY exhibited complete inhibition of HuNoV GII.4 replication (genomic copies at or below the lower limit of detection of 200 copies/reaction) in J2 HIE cells at dilutions of up to 1:32 (0.3125) mg/mL) for isolate AB2, and 1:64 (0.1563 mg/mL) for isolate 3241SK. A significant reduction in genome copies compared with unimmunized IgY was seen at 1:64 in isolate AB2, and 1:128 in isolate SK3241. In contrast, no neutralization with Unimmunized IgY was observed (Figure 2). These results indicate a within-genotype cross-variant reactivity of the anti-HuNoV IgY, generated against the 1974 CHDC 2094 strain VLP with two isolates of the currently circulating 2012 GII.4 [P16] Sydney strain.

To examine cross-genotype reactivity of anti-GII.4 HuNoV IgY, we examined its neutralization activity against the GII.6 3650SK virus. Serial 2-fold dilutions of anti-GII.4 HuNoVLP IgY were tested. The average fold change for the GII.6 viral genome (1–24 h) was 1.59. In this experiment, detectable neutralization was not observed against HuNoV GII.6 at the highest concentration tested (1:2; 5 mg/mL) (Appendix A). Of note, replication of HuNoV GII.6 in J2 HIE cells was approximately two log_10_ lower than HuNoV GII.4. Genome copies of HuNoV GII.4 viruses at 24 h were approximately 10^6^ genome copies per well, while the genome copies of HuNoV GII.6 at 24 h were approximately 10^4^ genomic copies per well. In all J2 HIE assays, 2′-CMC at a concentration of 200 µM resulted in complete inhibition of either GII.4 or GII.6 virus replication. CV values for each experiment are shown in Appendix A. Taken together, these findings demonstrate that anti-GII.4 VLP IgY can neutralize viral replication of live GII.4 viruses in vitro. Notably, neutralization occurred against GII.4 viruses that differ from that used to generate the IgY, suggesting that within-genotype cross-reactivity occurs. In contrast, a lack of neutralization against GII.6 suggests that across-genotype cross-reactivity is not observed, at least not between GII.4 and GII.6.

## 4. Discussion

This study demonstrates that polyclonal, HuNoV-specific IgY can be produced at high and sustained titers for at least nine months following initial hen immunization with HuNoVLP in combination with a poultry-specific adjuvant (S3). Furthermore, anti-HuNoV IgY effectively blocked the essential viral interaction with histo-blood group antigens required for cell entry and infection, at concentrations as low as ca. 0.005 mg/mL (1:2048, Figure 1). HBGA-blocking assays have previously been demonstrated to be highly correlated with assays that assess HuNoV neutralization [40]. Previous work by our group has shown that IgY raised against viral adhesion molecules, i.e., spike glycoprotein of SARS-CoV-2, effectively reduces receptor binding and viral replication in vitro [26]. These findings, taken together with the study reported here, support the broad potential applicability of this therapeutic approach.

While the anti-adhesion effect of anti-HuNoV IgY was demonstrated against VLP created from the capsid protein of an older HuNoV isolate, GII.4 CHDC2094/1974/US, which was also used to immunize the IgY-producing laying hens, we demonstrated viral neutralization by that IgY of a much more recent HuNoV strain, GII.4[P16] Sydney 2012 [41]. The VP1 capsid protein of the 1974 immunizing strain has known sequence heterology (95% aa homology) with the newer Sydney strain [42], demonstrating within-genotype cross-variant reactivity. This is one of the potential benefits of polyclonal, compared with monoclonal, antibodies (mAbs), which fail to bind to within-genotype variants [3,43]. However, no between genotype cross-reactivity was seen in this study between GII.4-directed IgY and GII.6 live HuNoV, suggesting that clinically useful IgY preparations will need to be prepared against each active circulating HuNoV genotype to be covered. Recent work demonstrates that HuNoV P domain complexes (“P-particles”) can also be used as vaccine platforms, potentially permitting the creation of broadly reactive IgY against multivalent HuNoV immunogens [13].

Importantly, we showed significant inhibition of HBGA binding to HuNoVLP by unimmunized IgY, albeit at only a relatively high concentration of IgY. However, we found that unimmunized IgY failed to neutralize HuNoV in the HIE model, highlighting the importance of performing neutralization studies and not relying exclusively on HBGA blocking assays when testing potential HuNoV therapeutic candidates.

Partial therapeutic effects have previously been reported for unimmunized IgY in animal studies of human rotavirus disease. For example, Vega et al., 2012 demonstrated a small but significant reduction in diarrheal disease severity by unimmunized IgY used as a control group in gnotobiotic pigs challenged with human rotavirus [44]. Nonetheless, that study also showed significantly greater therapeutic effects of immunized rotavirus-specific IgY by comparison with unimmunized IgY. We believe that these findings, taken together, suggest a low-level, nonspecific blocking capacity by non-specifically immunized polyclonal IgY, without therapeutic implications for that material.

Others have previously reported the production of anti-HuNoV IgY. Dai et al. used HuNoV “P-particles,” which represent the protruding domain of the capsid protein, as the immunogen, using Freund’s complete and incomplete adjuvant for primary and two booster injections, respectively, and reporting IgY titers in yolks ranging from 1:200,000 to 1:500,000, with successful blocking of HBGA receptors [45]. Zhu et al. used a recombinant vesicular stomatitis viral vector to induce production of HuNoV VP1 protein in eggs, reporting “high titers” and again demonstrating HBGA blocking by the resulting IgY [15]. Here, we report the use of HuNoV GII.4 CHDC2094 VLP as the immunogen, administered with a designated poultry adjuvant, Montanide ISO 70 VG, and attaining sustained titers of greater than 1:1.2 million, at least two-fold greater than that reported by Dai [45]. Use of the Montanide adjuvant is an important contribution to the “three R’s” of animal welfare; hens immunized with Montanide ISA 70 VG have been shown to have fewer and less severe instances of tissue damage at injection sites compared with those receiving other adjuvants such as Freund’s complete or incomplete adjuvant [46]. To our knowledge, this study is the first report to demonstrate IgY-mediated viral neutralization of noroviruses in HIE model, and the first report of substantial cross-variant reactivity of IgY within the GII.4 genotype.

Passive immunization by direct delivery of neutralizing antibodies in at-risk individuals has demonstrated the validity of the approach, while also highlighting existing limitations. Previous work has shown, for example, that bovine IgG antibodies directed against a leading bacterial pathogen, enterotoxigenic *Escherichia coli* (ETEC), delivered in bovine colostrum, can reduce incidence and volume of diarrheal stools in human volunteers challenged with a single strain of ETEC (H10407), and at least one commercial product employing bovine IgG against ETEC is available [47,48]. 

Treatment of chronic HuNoV infections with enterally administered pooled human immunoglobulins has recently been reported in two small case series, with complete resolution of symptoms in more than 90% of observed cases [49,50]. While these findings support the use of exogenous anti-HuNoV antibodies, the costs of administering this material were prohibitive (up to about $600 per treatment), and a massive scale-up of this approach is impractical [49].

Various mAbs have been developed for oral administration to target a range of human disorders [46,51]. mAbs, however, lack the relatively broad epitope recognition provided by polyclonal antibodies, even failing to cross-react with HuNoV capsids within the GII.4 genotype [52], whereas we have shown good within-genotype reactivity with polyclonal IgY. While mAb production has recently been dramatically scaled up during the COVID-19 pandemic, reducing their cost substantially, these drugs are indicated only for treatment and post-exposure prophylaxis, and only by intravenous injection [53]. For a typical traveler or immune-compromised person at risk for chronic HuNoV infection, however, pre-exposure prophylaxis, not treatment, is likely to be the therapeutic goal. For these purposes an orally administered therapeutic is likely to be an important contributor to uptake and use. The demonstrated effectiveness or orally administered IgY in human rotavirus and *Helicobacter pylori* disease suggests the practicality of the oral route for these antibodies [20,21,54,55]. To date, no orally administered mAbs have been approved for use [51]. 

The easy and speedy IgY production suggest that new IgY can readily be produced against non-cross-reacting variants to enhance prophylactic coverage. Each hyperimmunized raw egg contains roughly 100 mg total polyclonal IgY, making large-scale, efficient production possible at a low cost [17]. Polyclonal antibodies have been shown to have broader neutralizing properties than mAbs against some viral pathogens, which may be advantageous in the face of antigenic diversity among pathogen strains [52]. Furthermore, IgY fails to fix mammalian complement and does not bind to mammalian F_c_ receptors, and potentially reduces the risk of undesired complement activation and inflammation [56,57]. Target-specific IgY can be developed quickly at low cost by immunization of a small number of hens, permitting rapid iteration of target antigens to select a final optimized form [17]. Studies in agricultural and laboratory settings suggest that IgY targeting animal enteropathogens is effective at both prevention and mitigation of diarrheal symptoms [22]. It is also worth noting that Chen et al., 2022 recently proposed a simplified protocol for IgY extraction without chemical excipients or high-speed centrifugation [58], which may facilitate future commercial applications. 

Valid questions remain about the survivability of active IgY in the human upper digestive tract, where acid and pepsin rapidly degrade unprotected IgY [59]. Prior study of orally administered anti-rotavirus (HRV) IgY in a matrix of cow milk in a gnotobiotic pig model demonstrated full protection against HRV-associated diarrhea and significantly reduced virus shedding [44], suggesting that similar protection will be conferred by anti-HuNoV IgY. Further study of anti-HuNoV IgY is planned in a gnotobiotic pig model of HuNoV infection and diarrhea, which recapitulates HuNoV biology with an oral route of infection and subsequent diarrhea, transient viremia, and virus shedding in feces [60,61,62]. For more general clinical use in humans, systems for microencapsulation of IgY for oral administration have been described previously and shown to protect activity of IgY both in a pig model and in simulated gastric fluid [59,63,64].

Additional evidence for practicality of IgY for human use comes from a recent human study of anti-*Helicobacter pylori* IgY demonstrating the survival of 78% of specific IgY activity in simulated gastric fluid (SGF) after 30 min of exposure when administered 1/16 (*w/w*) with powdered skim milk, a 56% reduction in breath hydrogen levels, and an overall reduction of symptoms of 87% following a 14-day course of treatment [21]. Further human evidence comes from a 2019 systematic review and meta-analysis of infants receiving anti-rotavirus IgY as treatment for rotavirus diarrhea (17 randomized clinical trials, 1347 received IgY and 1279 received conventional treatment; subjects receiving IgY treatment were significantly (*p* < 0.0001) more likely to be “effectively treated” than those receiving conventional care (OR = 3.87) [55]. It was recently shown that enteric viruses, including HuNoV, can establish infection in salivary glands of mouse pups [65]. While further validation in humans is required, this observation could suggest an alternative site of therapeutic action for orally administered antiviral IgY.

It is worth noting that the present study has certain limitations. The number of hens used to generate HuNoV IgY was small, which has a potential impact on generalizability. There is a small possibility of inadvertent inclusions of potential pathogens that could be transmitted to human subjects as a result of using ordinary agricultural laying hens, and work is underway to elucidate detection or prevention (e.g., through use of specific pathogen free hens) of pathogen cross over in the IgY material collected. Furthermore, this study did not include any in vivo evaluations, limiting the strength and applicability of its conclusions, though the HIE results are encouraging. While resources permitted only examination of anti-GII.4 IgY against GII.4 and GII.6 HuNoV genotypes, future studies will examine IgY targeting additional strains and genotypes of these pathogens, and subsequently in relevant animal models such as gnotobiotic pigs [66].

## 5. Conclusions

In this study, avian IgY were produced against GII.4 HuNoVLP at high and sustained titers. HuNoVLP IgY was effective at disrupting HBGA adhesion of HuNoVLP and reduced HuNoV replication in an HIE model, even when the immunizing strain and the active viral strains used in HIE differ with regard to capsid protein sequences. Large-scale production of anti-HuNoV IgY may be a rational approach to prophylaxis and treatment of HuNoV infection. Passive oral immunization with target-specific IgY may provide an appealing approach until vaccines are developed with the desired efficacy and spectrum of activity and, even once such active prophylaxis is available, may provide a useful alternative in patients unable or reluctant to receive vaccines.

## Figures and Tables

**Figure 1 viruses-14-02371-f001:**
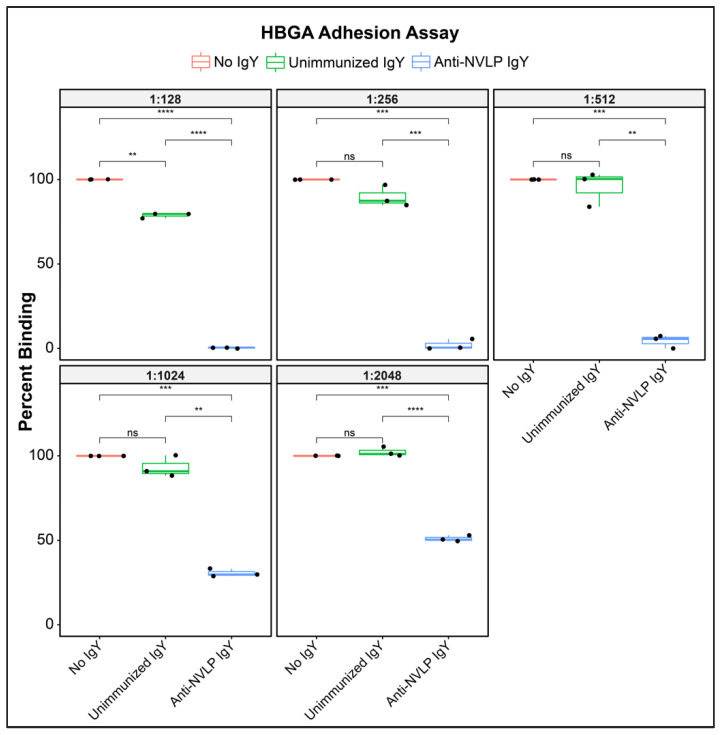
Box and whisker plots of inhibition of HuNoVLP adhesion to HBGA antigens in a cell-free system. Boxes represent interquartile range (IQR) with median, shown as center bar of each sample group (N = 3 replicates per group). Whiskers represent 1.5 times the IQR. *p*-value, by two-sample *t*-test method, and 95% confidence interval (CI) was calculated using R software package EnvStats (v.2.3.1) [28]. (ns), not significant, *p* > 0.05; (**), *p* ≤ 0.01; (***), *p* ≤ 0.001; (****), *p* ≤ 0.0001.

**Figure 2 viruses-14-02371-f002:**
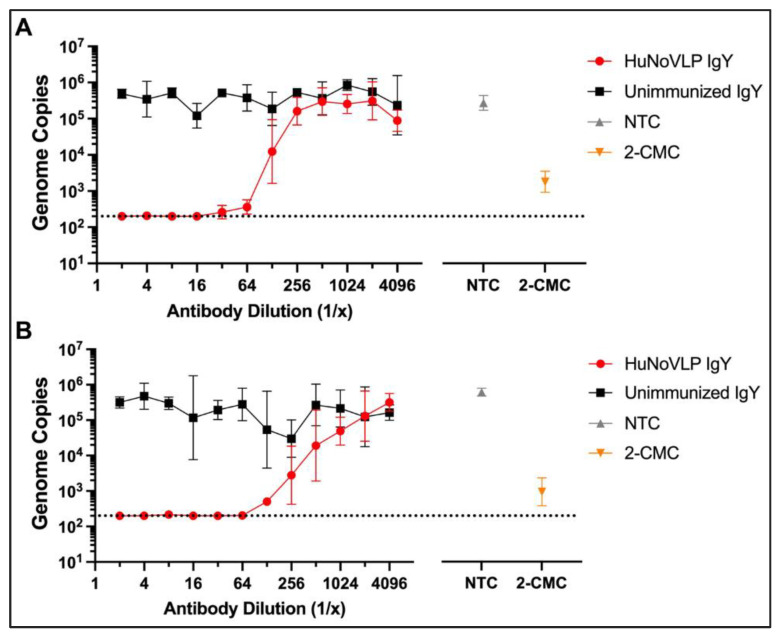
Comparison of neutralization activities of immunized (HuNoVLP IgY) and unimmunized IgY against two isolates (**A**) AB2 and (**B**) 3241SK of HuNoV GII.4 in J2 HIE cells (geometric mean with 95% confidence intervals) showing pooled results of two independent experiments (N = 8 total replicates per treatment condition). Antibodies were prepared in serial 2-fold dilutions. 2-CMC (2′-C-methylcytidine), a small-molecule viral polymerase inhibitor [39] was used as a positive drug control and tested at a concentration of 200 µM. NTC: Non-treatment control for virus growth in the cells (medium alone + virus). Dashed line indicates RT-qPCR limit of detection (2 × 10^2^ genome copies per reaction).

**Table 1 viruses-14-02371-t001:** Primers and probes used in this study. ^a^, Parentheses indicate polarity; ^b^, Y = C/T, R = A/G, I = deoxyinosine, N = A/C/T/G, H = A/C/T; ^c^, Miura et al., (2013) [35]; ^d^, Cannon et al., (2017) [36].

Genogroup	Sequence Name ^a^	Sequence 5′–3′ ^b^
GII.4 ^c^	QNIFS (probe)	/56-FAM/AGC ACG TGG /ZEN/GAG GGC GAT CG/3IABkFQ/
GII.4 ^c^	QNIF2d (+)	ATG TTC AGR TGG ATG AGR TTC TCW GA
GII.4 ^c^	COG2R (-)	TCG ACG CCA TCT TCA TTC ACA
GI ^d^	MON 432 (+)	TGG ACI CGY GGI CCY AAY CA
GI ^d^	G1SKR (-)	CCA ACC CAR CCA TTR TAC A
GII ^d^	MON 431 (+)	TGG ACI AGR GGI CCY AAY CA
GII ^d^	G2SKR (-)	CCR CCN GCA TRH CCR TTR TAC AT

## Data Availability

The data presented in this study are available in the present article and Appendix A.

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
