# Peer review of "Feasibility of Polyclonal Avian Immunoglobulins (IgY) as Prophylaxis against Human Norovirus Infection"

_viruses, 2022, doi:10.3390/v14112371_

Round 1

Reviewer 1 Report

The manuscript by Artman et al. describes the generation of anti-HuNoV VLP IgY and its characterization in HBGA inhibition and HIE neutralization assays. While some anti-HuNoV IgY preparations have been described in the past, this is the first study testing neutralization activity of the serum in the enteroid system. The data are of general interest to the readership. However, numerous weaknesses were noted in the manuscript that need to be corrected to improve the readability and rigor of the manuscript as outlined below.

Major weaknesses:

1)    The readability of the manuscript would be greatly improved by including introductory sentences and concluding sentences for each subheading in the results section so the reader has a context why a particular experiment was done and what the take-home message is.

2)    Scientific rigor of data is unclear. Figure legends do not consistently indicate how many independent experiments were performed to derive data. The statistical assay used for data analysis also needs to be included.

3)    Line 213: Methods state that TCID50 was used to quantify infectivity of filtrates. However, HuNoV does not cause CPE in HIE, therefore it is not possible to perform a classical TCID50 assay (or plaque assay). Please clarify the method used to measure infectivity.

4)    Figure 1 is missing a data point from a very high dilution that does not show effective inhibition. This demonstrates the dose-responsiveness of the assay.

Minor weaknesses:

1)    Line 183: Methods state that binding inhibition was calculated but Figure 1 shows percent binding. Please clarify.

2)    Line 272: Text states: “This material was used in all subsequent analyses.” However, it is unclear exactly which material is being used since Figure S4 shows purifications from 4 weeks post-inoculation and a preparation from 7-11 weeks postimmunization. Please clarify which specific preparation is being used throughout. If preparations from different timepoints are being used, please specify the material used for each experiment.

3)    Lines 301-303: Text states: “HuNoVLP IgY exhibited complete 301 inhibition of HuNoV GII.4 replication in HIE J2 cells at a dilution of 1:32 (0.313 mg/mL) and 1:128 (0.078 mg/mL)” However, data in Figure 2 shows partial neutralization in experiment A at 1:64 and 1:128 and complete neutralization at 1: 1:256, while in experiment B partial neutralization was observed at 1:256 and full neutralization at 1: 512. Therefore, the conclusions need to be modified.

4)    Lines 353-354: indicates significant inhibition of HBGA binding by unimmunized IgY. While a statically significant difference was observed at low dilutions in Figure 2, there was no neutralizing capacity of the unimmunized serum seen in Figure 2. Therefore, the conclusion that unimmunized IgY can be used as prophylaxis (line 358) is fundamentally flawed and this section of the discussion needs to be rephrased. This finding however highlights the importance of performing neutralization studies and not overly relying on HBGA blocking assays when testing potential anti-HuNoV treatment modalities.

5)    Line 386: what is “cHuNoV”? Please ensure all abbreviations are defined at first use throughout and if only used rarely, please refrain from using abbreviations and spell out.

6)    Line 396-397: states that production of mAbs is expensive and inefficient. However, the COVID-19 pandemic has clearly shown that mAbs (e.g. bamlanivimab) can be successfully used to treat virus infections and that the production of these preparations can be scaled up sufficiently for human patient use. Therefore, the statement should be modified.

7)    Line 438: states that data show minimal difference in IgY between hens. However, no data is provided, which compares the efficacy of sera from different animals. Please provide such data or remove statement.

8)    Line 456: states that IgY provide a useful alternative in patient unable or reluctant to receive vaccines. However, no vaccines are available to prevent HuNoV infections. Therefore, the statement needs to be modified.

9)    Figure 2: Since A and B are repeats of the same experiment, they can be shown in the same graph (or at least should be shown on the same scale). If there are differences between A and B that do not allow one to combine the data, please provide such information.

10) Figure S1: the graph shows 2CMC response curve from which EC50 was calculated. However, a CC50 value is also stated. Please also provide the data for cell viability that was used to calculate CC50.

11) Figure S3 legend: states “Immunizations were given at weeks 0, 2, and 4 (arrows)” but no arrows are shown. Furthermore, 2-21 represents a titer of 1: 2,097,152, not 2,097,152. Please correct. Lastly, what is the limit of detection of the ELISA assay? Please add info to legend.

12) Figure S5: No data point for 2CMC is shown. Also, generation of “anti-RBD (receptor binding domain) IgY” is not described in the manuscript. Please include that information in methods and results or remove data.

13) Figures S5 and S6: please clarify in the legend what the dashed line represents

Author Response

Comments and Suggestions for Authors

The manuscript by Artman et al. describes the generation of anti-HuNoV VLP IgY and its characterization in HBGA inhibition and HIE neutralization assays. While some anti-HuNoV IgY preparations have been described in the past, this is the first study testing neutralization activity of the serum in the enteroid system. The data are of general interest to the readership. However, numerous weaknesses were noted in the manuscript that need to be corrected to improve the readability and rigor of the manuscript as outlined below.

Major weaknesses:

  • The readability of the manuscript would be greatly improved by including introductory sentences and concluding sentences for each subheading in the results section so the reader has a context why a particular experiment was done and what the take-home message is.

Au Response: We thank the Reviewer for pointing out how we could improve clarity and readability, and have included introductory and concluding sentences in the Results section.

  • Scientific rigor of data is unclear. Figure legends do not consistently indicate how many independent experiments were performed to derive data. The statistical assay used for data analysis also needs to be included. 

Au Response: We appreciate the Reviewer pointing out these vital omissions. We have now indicated the number of independent experiments, as well as the number of replicates in each group (“N”). The statistical assay used for data analysis is now included in the relevant Figure captions and is referenced.

3)    Line 213: Methods state that TCID50 was used to quantify infectivity of filtrates. However, HuNoV does not cause CPE in HIE, therefore it is not possible to perform a classical TCID50 assay (or plaque assay). Please clarify the method used to measure infectivity. 

Au Response: Thank-you to the reviewer for pointing out the lack of description in regard to how the TCID50 values are calculated for noroviruses, where CPE or plaques are not observed. This was performed as previously described by Costantini et al. 2018 (now referenced) whereby an increase in viral genomes as measured by RT-qPCR serves as the readout for whether a well is positive (a >10-fold increase in copies) or negative (no increase in viral RNA from time zero), with the resulting data used to determine the TCID50 by the classical Spearman-Karber method. While this method is admittedly atypical for calculating TCID50, it is still reporting the dose of virus that is required to infect 50% of the wells and appears to be robust at doing so. Until a plaque, focus-forming, or CPE based assay is developed, this currently appears to be the only method to do this. We have added a more complete description of the technique in the Methods section, with a brief explanation of why we selected the Costantini approach, given the inapplicability of standard CPE/plaque assay methodology.

4)    Figure 1 is missing a data point from a very high dilution that does not show effective inhibition. This demonstrates the dose-responsiveness of the assay.

Au Response: We agree and thank the reviewer for pointing this out. The primary reason for carrying out the HBGA assay was to screen for anti-adhesion effects, as a means of qualifying the selected IgY as a candidate for use in the viral neutralization studies in HIE. Hence, we did not carry the antibody dilutions out to complete loss of inhibition, using 1:2048 as the highest antibody dilution. We note that there is a small but evident reduction in HBGA adhesion at the higher dilutions, and that this reduction is significant (p<0.05) by comparison with the lower dilutions (p=0.000017 for 2048 vs. 128; 0.00123 for 1024 vs. 128; and 0.0031 for 1024 vs. 512. As is apparent from visual inspection of the Figure, the effect of dilution from 128 to 512 is non-significant.  We have included a brief explanation of these findings in the text just above the Figure itself.

Minor weaknesses:

  • Line 183: Methods state that binding inhibition was calculated but Figure 1 shows percent binding. Please clarify.

Au Response:  We thank the Reviewer for noting this discrepancy and have changed the language in the Methods section accordingly.

  • Line 272: Text states: “This material was used in all subsequent analyses.” However, it is unclear exactly which material is being used since Figure S4 shows purifications from 4 weeks post-inoculation and a preparation from 7-11 weeks postimmunization. Please clarify which specific preparation is being used throughout. If preparations from different timepoints are being used, please specify the material used for each experiment.

Au Response: This was indeed unclear, and we appreciate the Reviewer’s calling this to our attention. We used material from pooled yolks collected during weeks 7-11, as shown in S4, for all analyses. We have indicated this at the new Line 301.

  • Lines 301-303: Text states: “HuNoVLP IgY exhibited complete 301 inhibition of HuNoV GII.4 replication in HIE J2 cells at a dilution of 1:32 (0.313 mg/mL) and 1:128 (0.078 mg/mL)” However, data in Figure 2 shows partial neutralization in experiment A at 1:64 and 1:128 and complete neutralization at 1: 1:256, while in experiment B partial neutralization was observed at 1:256 and full neutralization at 1: 512. Therefore, the conclusions need to be modified.

Au Response: We appreciate the Reviewer’s comment here and agree that this could have been improved by better descriptive language. From an abundance of caution, we used the term “Complete Inhibition” to indicate a viral copy number at or below the lower limit of detection, indicated by the dashed line at 200 genome copies. The Reviewer is of course correct in pointing out that significant inhibitory effects were demonstrated at some higher IgY dilutions, which we had not pointed out. We have inserted a definition of “complete inhibition” as used herein at Line 340. We have modified the text accompanying Figure 2 to indicate that we did see significant inhibition vs. unimmunized IgY at several higher dilutions.  One caveat we do not see “complete neutralization at 1:256” nor at 1:512, under our understanding of neutralization. At these dilutions, there is no significant difference in genome copies between the active and Unimmunized IgY groups (as indicated by the overlapping 95% confidence intervals).

  • Lines 353-354: indicates significant inhibition of HBGA binding by unimmunized IgY. While a statically significant difference was observed at low dilutions in Figure 2, there was no neutralizing capacity of the unimmunized serum seen in Figure 2. Therefore, the conclusion that unimmunized IgY can be used as prophylaxis (line 358) is fundamentally flawed and this section of the discussion needs to be rephrased. This finding however highlights the importance of performing neutralization studies and not overly relying on HBGA blocking assays when testing potential anti-HuNoV treatment modalities.

Au Response: Thank you to the reviewer for pointing out this obvious misstatement. The sentence appears to be an artifact of editing multiple versions. We agree entirely that nothing in any of our data suggests that unimmunized IgY could be used in prophylaxis and have removed the incorrect sentence. Additionally, we have now included a sentence emphasizing the Reviewer’s point about potential pitfalls of over-reliance on HBGA blocking assays when evaluating HuNoV therapeutic candidates.  We have also re-worded and expanded on the discussion of a minor impact of unimmunized IgY in a previous rotavirus study in gnotobiotic pigs, to further emphasize the point that we see no therapeutic role for this material.

  • Line 386: what is “cHuNoV”? Please ensure all abbreviations are defined at first use throughout and if only used rarely, please refrain from using abbreviations and spell out.

Au Response: Thank you for pointing this out. “cHuNoV” means “chronic HuNoV. We have written this out now and removed the original definition early in the introduction. This is a rare abbreviation and should not have been used.

  • Line 396-397: states that production of mAbs is expensive and inefficient. However, the COVID-19 pandemic has clearly shown that mAbs (e.g. bamlanivimab) can be successfully used to treat virus infections and that the production of these preparations can be scaled up sufficiently for human patient use. Therefore, the statement should be modified.

Au Response: This is an important observation, and we thank the Reviewer for it. We have removed the statement about expense and inefficiency of production for mAbs. We have also added several sentences pointing out the current roles for mAbs as treatment and post-exposure prophylaxis, contrasting those intended uses from the pre-exposure prophylactic use we envision for people interested in preventing HuNoV infection during travel, or when immunocompromised.   Finally, we have moved the sentence on simplified, inexpensive IgY extraction referencing Chen 2022 to the end of the succeeding paragraph so that it appears in the correct context.

  • Line 438: states that data show minimal difference in IgY between hens. However, no data is provided, which compares the efficacy of sera from different animals. Please provide such data or remove statement. 

Au Response: We appreciate this observation. While we have hen-to-hen comparison data for our work on other therapeutic IgY, we lack such data for anti-HuNoV IgY, and have removed the sentence as recommended.  

  • Line 456: states that IgY provide a useful alternative in patient unable or reluctant to receive vaccines. However, no vaccines are available to prevent HuNoV infections. Therefore, the statement needs to be modified.

Au Response: Thank you for noting this inconsistency. We have added a short phrase in the sentence to indicate that, even once vaccines become available, passive immune prophylaxis may be appealing to such people.

9)    Figure 2: Since A and B are repeats of the same experiment, they can be shown in the same graph (or at least should be shown on the same scale). If there are differences between A and B that do not allow one to combine the data, please provide such information.

      Au Response: The Reviewer makes an excellent point here that we believe improves the presentation of the data and the reader’s ability to interpret them. In our original submission, we failed to clearly identify that the experiments represented in panels A and B are in fact two separate experiments. In A, the GII.4 HuNoV was an isolate from a patient in Alberta (AB2), while in panel B the isolate was from a patient in Saskatchewan (SK3241). Hence these were not repeats of the same experiment and should not be shown on the same graph. We entirely agree that the two graphs should be shown on the same scale, and we have revised the panels in Figure 2 accordingly. The error bars here are helpful in discerning the extent of inhibition by active IgY at each dilution with that produced by Unimmunized IgY. As noted above, we have modified the text in the Results section to state that we observed complete inhibition (defined as viral copy numbers at or below the Lower Limit of Detection (LLOD) at dilutions from 1:2 through 1:16, and significant inhibition vs. Unimmunized IgY at dilutions from 1:32 to 1:128. The overlapping error bars at 1:256 and higher dilutions indicate a lack of significant differences at these dilutions.

  • Figure S1: the graph shows 2CMC response curve from which EC50 was calculated. However, a CC50 value is also stated. Please also provide the data for cell viability that was used to calculate CC50.

Au Response: Thank you for this opportunity to clarify our intentions. The curve shown in S1 was used only to establish a reasonable dose to use for 2CMC as the positive control. The CC50 value shown is “greater than” 400 micromolar, i.e., there were no cytopathic effects of 2CMC at or below this concentration, which was the highest tested. The selected concentration of 200 micromolar is well below this concentration while still more than twice the EC50, indicating that 2CMC has good antiviral effects without risk of drug-induced cytotoxicity in the assay at 200 micromolar.

  • Figure S3 legend: states “Immunizations were given at weeks 0, 2, and 4 (arrows)” but no arrows are shown. Furthermore, 2-21represents a titer of 1: 2,097,152, not 2,097,152. Please correct. Lastly, what is the limit of detection of the ELISA assay? Please add info to legend. 

Au Response: We appreciate these important points. The S3 image shows boxes around the week numbers when immunizations were given; the legend incorrectly indicated that these were arrows. We have corrected the legend. We have also included the correct representation of the titer (now “1:2,097,152”), and added language about the LOD, while indicating that the lower limit of detection (LLOD) for this assay was 2.1 times the OD450 value for the unimmunized control and defined as zero, as indicated in Methods at the end of the section on ELISA titers of IgY in HuNoVLP-immunized hens’ eggs

Figure S5: No data point for 2CMC is shown.

Au Response: Thank you for these observations. We note that Figure S5 does indicate a single data point for HIE Infected with HuNoV and treated with 2CMC, as a green square at just below a genomic equivalent of 104. The generation of anti-RBD IgY is described under “Anti-HuNoV activity of IgY antibodies on HIE cells” in Methods at roughly line 248, with a reference to a 2016 publication by Carmona-Vicente.   

  • Figures S5 and S6: please clarify in the legend what the dashed line represents

Au Response: We appreciate this obviously important point. The dashed line indicates lower level of detection for HuNoV viral copies (200 genomic equivalents). This information has been added to the captions for Figures S5 and S6.

Reviewer 2 Report

In the manuscript by Artman et al., entitled " Feasibility of Polyclonal Avian Immunoglobulins (IgY) as Prophylaxis 2 Against Human Norovirus Infection", the authors present an interesting effect of avian HuNoV-specific IgY, preventing HuNoV infection in vitro in a human intestinal enteroid (HIE). Through immunization of Hens with Human Norovirus-Like Particles, the authors show that this approach can get high titers of HuNoV-specific IgY. The authors further characterized the HuNoV-specific IgY antiviral effect and demonstrated that HuNoVLP IgY inhibited virus replication in HIE by effectively blocked the viral interaction with histo-blood group antigens which required for human norovirus entry. The manuscript is logically presentedand interpretations are supported by the data presented. My concerns are few and highlighted below. 
1. Please use high resolution figure for figure1. 

2. The authors should provide more information for Fig S2.

3. Please check throughout for the format of citation.

Author Response

In the manuscript by Artman et al., entitled " Feasibility of Polyclonal Avian Immunoglobulins (IgY) as Prophylaxis 2 Against Human Norovirus Infection", the authors present an interesting effect of avian HuNoV-specific IgY, preventing HuNoV infection in vitro in a human intestinal enteroid (HIE). Through immunization of Hens with Human Norovirus-Like Particles, the authors show that this approach can get high titers of HuNoV-specific IgY. The authors further characterized the HuNoV-specific IgY antiviral effect and demonstrated that HuNoVLP IgY inhibited virus replication in HIE by effectively blocked the viral interaction with histo-blood group antigens which required for human norovirus entry. The manuscript is logically presented, and interpretations are supported by the data presented. My concerns are few and highlighted below. 
1. Please use high resolution figure for figure1. 

Au Response: Thank you for pointing out the lower-quality image that was used as a placeholder. We have replaced that with a high-resolution file.

  1. The authors should provide more information for Fig S2. What was the reference material for Prepare HOV36 RNA Transcript (GII.4 RNA standard)?

Au Response: Thank you for the opportunity to clarify this. We have added a paragraph on the Methods used to prepare this Figure in the Supplementary Material, and included the reference to the method selected (Schwab 1997).  

  1. Please check throughout for the format of citation.

Au Response: Thank you. We have added new citations and reformatted the entire bibliography. We believe the format is now consistent throughout the manuscript.

Reviewer 3 Report

The authors proposed a novel idea to use avian IgY to prevent human norovirus infection. It’s very interesting. The results look encouraging, although the high dilution of IgY doesn’t work well. Moreover, the authors fully discuss in the manuscript the promising of the application of avian IgY.

My question is if any previous publication shows the application of avian IgY to prevent virus infection. I don’t see any introduction about this.

There is still some minor part can be improved. For Figure 2, the curve should be a curve of logistic regression. The authors should calculate the IC50 from the data.

And the authors don’t have any animal experiment (i.e. test in mice) in the study, which makes the conclusion not so strong. Can the authors add a testof the avian IgY against the infection of norovirus on mice?

Author Response

The authors proposed a novel idea to use avian IgY to prevent human norovirus infection. It’s very interesting. The results look encouraging, although the high dilution of IgY doesn’t work well. Moreover, the authors fully discuss in the manuscript the promising of the application of avian IgY.

My question is if any previous publication shows the application of avian IgY to prevent virus infection. I don’t see any introduction about this.

Au Response: Thank you for inquiring about this. While we had included some description of prior use of IgY in the Discussion, we failed to indicate this in the Introduction, and have now added a brief paragraph near the end of the Introduction.

There is still some minor part can be improved. For Figure 2, the curve should be a curve of logistic regression. The authors should calculate the IC50 from the data. 

Au Response: Thank you. We assume the Reviewer refers to Figure S2, not the Figure 2 in the main manuscript, since these data cannot be represented by a curve of logistic regression. We have included the correct logistic regression curve in Figure S2. We note that this is purely the standard curve for RT-qPCR using a plasmid to allow copy numbers to be calculated, hence, there is no inhibition present, so there is no way to calculate IC50 from these data. We have also added a paragraph in the S2 caption to indicate how the curve was generated. 

And the authors don’t have any animal experiment (i.e. test in mice) in the study, which makes the conclusion not so strong. Can the authors add a test of the avian IgY against the infection of norovirus on mice?

Au Response: The Reviewer raises a critical point, however, to date human norovirus has not been successfully modeled in mice. We point out that we are planning a study in gnotobiotic pigs, for which a valid infection model has been established, and that this is described in brief in the Discussion section, with references to the pig model.

Round 2

Reviewer 1 Report

The authors nicely and thoroughly addressed the previous critiques. No further action is needed in this reviewers opinion.